# Predictive value of uric acid to albumin ratio for carotid atherosclerosis in type 2 diabetes mellitus: A retrospective study

**Yao Yin[1], Liyin Zhang[1], Jiaoyue Zhang[2]\*, Si Jin [1]\***

1 Department of Endocrinology, Institute of Geriatric Medicine, Liyuan Hospital, Tongji Medical College, Huazhong University of Science and Technology, Wuhan, China, 2 Department of Endocrinology, Union Hospital, Tongji Medical College, Huazhong University of Science and Technology, Wuhan, China

\* JinSi@hust.edu.cn (SJ); zhangjiaoyue@126.com (JZ)

## Abstract

### Background

This study aims to evaluate the correlation between the uric acid (UA) to albumin (ALB) ratio (UAR) and carotid atherosclerosis (CAS) in patients with type 2 diabetes mellitus (T2DM), as well as to assess the predictive value of UAR for CAS.

### Methods

A cross-sectional, single-center study was conducted, retrospectively analyzing hematological parameters from 259 T2DM patients with CAS (T2DM-CAS) and 131 T2DM patients without CAS (T2DM-WCAS). Carotid intima-media thickness (IMT) and carotid plaques (CAP) were measured using Doppler ultrasound.

### Results

The UAR level in the T2DM-CAS group was significantly higher than that in the T2DM-WCAS group (P < 0.001). Multivariate logistic regression analysis revealed that UAR is an independent risk factor for T2DM-CAS (P < 0.001). The area under the ROC curve (AUC) for UAR in predicting T2DM-CAS was 0.712, with a Youden index of 0.278.

### Conclusion

High levels of UAR are closely associated with the occurrence of T2DM-CAS and may serve as a useful biomarker for predicting T2DM-CAS.

## 1. Introduction

Diabetes is a common yet potentially devastating disease, with rising prevalence over the past few decades, posing a significant public health challenge in the 21st century [1]. It imposes a substantial burden of disease on communities in both developed and developing countries, and its associated complications—both macrovascular and microvascular—have a devastating

**Data availability statement:** All relevant data are within the paper and its Supporting Information files.

**Funding:** This work was supported by the National Natural Science Foundation of China (Grant No. 82070862, No.82370840) and the National Key R&D Program of China (Grant No. 2020YFC2008900). These funds were obtained by Professor Jin.

**Competing interests:** The authors have declared that no competing interests exist.

**Abbreviations:** AS, Atherosclerosis; CAS, Carotid atherosclerosis; CIMT, Carotid intima-media thickness; CIMTA, Abnormal carotid intima-media thickness; CAP, Carotid atheromatous plaque; CVD, Cardiovascular Disease; T2DM, Type 2 Diabetes Mellitus; UA, Uric acid; ALB, Albumin; UAR, Uric acid/ Albumin ratio; T2DM-CAS, Type 2 diabetes mellitus with carotid atherosclerosis; T2DM-WCAS, Type 2 diabetes mellitus without carotid atherosclerosis.

impact on diabetic patients [2]. The carotid artery serves as a "window" reflecting systemic atherosclerosis, with carotid atherosclerosis (CAS) referring to the presence of atherosclerotic disease within the carotid arteries [3]. CAS plays a crucial role in the development of cardiovascular and cerebrovascular diseases [4–6]. Type 2 diabetes mellitus (T2DM) has been identified as an independent risk factor for the accelerated progression of atherosclerosis [7], and atherosclerotic cardiovascular disease (ASCVD) is the leading cause of morbidity and mortality among T2DM patients [8]. Therefore, the early detection and treatment of CAS in individuals with T2DM (T2DM-CAS) are critical for preventing ASCVD and its related complications.

As the final product of purine metabolism, uric acid (UA) is associated with oxidative stress, inflammatory responses, and endothelial dysfunction [9]. Studies have indicated that elevated serum UA levels may serve as an independent risk factor for the progression of CAS and vascular events [10]. UA promotes the proliferation and migration of vascular smooth muscle cells (VSMCs) through the activation of MAPK pathways and oxidative stress, contributing to the progression of atherosclerotic plaques [11]. Serum albumin (ALB) is a marker of nutritional status and can bind to nearly all known endogenous compounds, metal ions, and exogenous substances, playing a vital role in the body's antioxidant defense [12]. Its antioxidant properties help prevent oxidative stress, a major factor contributing to endothelial dysfunction and the progression of atherosclerosis. ALB also plays a role in regulating endothelial function and vascular tone. A decrease in its levels can impair the bioavailability of nitric oxide, leading to vasoconstriction, increased vascular resistance, and impaired coronary perfusion, thereby making patients more susceptible to ischemic events [13]. Additionally, albumin-based indicators, such as the Naples Prognostic Score (NPS) and the Prognostic Nutritional Index (PNI), have demonstrated significant predictive value [14].

The uric acid to albumin ratio (UAR) is a novel biomarker that combines elements of nutrition, inflammation, and metabolic syndrome. Multiple studies have highlighted the close association between inflammation and the UAR. Research indicates that UAR is an independent risk factor for new-onset atrial fibrillation (AF) in patients with STEMI [15] and is also independently associated with the recurrence of AF after catheter ablation [16]. In STEMI patients undergoing percutaneous coronary intervention (PCI), UAR is independently related to no-reflow phenomena [17]. Additionally, UAR serves as an effective indicator of disease severity in patients with chronic coronary artery disease [18]. Furthermore, research suggests that when albumin is combined with UA, its prognostic value for peripheral arterial disease is enhanced [19]. These findings underscore the clinical significance of UAR as a potential biomarker in cardiovascular disease.

However, there is currently no research exploring the association between UAR and T2DM-CAS. Therefore, we hypothesize that UAR may serve as a predictive biomarker for T2DM-CAS. This study aims to explore the potential of UAR as a novel biomarker for the early screening and intervention of T2DM-CAS, which could contribute to reducing the risk of further vascular complications.

## 2. Materials and methods

### 2.1. Study population

This study enrolled 390 patients diagnosed with T2DM who were hospitalized in the Endocrinology Department at Union Hospital, Tongji Medical College, Huazhong University of Science and Technology, between January 1, 2023, and March 31, 2024. According to the 2019 European Society of Cardiology guidelines [20], T2DM was defined as a fasting blood glucose (FBG) level ≥7.0 mmol/L and/ or a glycated hemoglobin (HbA1c) level ≥6.5% and/ or

a 2-h plasma glucose ≥ 11.1 mmol/L during an oral glucose tolerance test (OGTT) and/ or a random plasma glucose ≥ 11.1 mmol/L. The inclusion criteria were patients aged 18-79 years with a confirmed diagnosis of T2DM. Our exclusion criteria were as follows: (1) Undefined or suspected non-type 2 diabetes mellitus; (2) Acute diabetic complications (such as diabetic ketoacidosis, hyperglycemia hyperosmotic state, and lactic acidosis); (3) Cardiovascular and cerebrovascular diseases, such as heart failure, coronary heart disease, history of stent implantation or acute cerebral infarction; (4) Confirmed liver cirrhosis with Child–Pugh C functional impairment or chronic kidney disease with eGFR below 60 mL/min; (5) Severely impaired consciousness or poor general condition; (6) History of hematological malignancies or active malignancies; (7) Thyroid dysfunction, autoimmune diseases, or chronic inflammatory conditions; (8) Confirmed pancreatic insufficiency, chronic pancreatitis or previous pancreatic surgery; (9) Use of immunosuppressants, glucocorticoids, or anticoagulants; (10) Leukocytosis ($> 10 \times 10^9$ cells/L) or leukopenia ($< 4 \times 10^9$ cells/L); (11) Incomplete clinical data. By excluding patients with non-type 2 diabetes, acute complications, and other severe comorbidities, we ensured the homogeneity of the study population. Additionally, the exclusion of patients with tumors, pancreatic diseases, and autoimmune disorders minimized the potential confounding effects of these conditions on metabolic parameters. These exclusion criteria were designed to enhance the accuracy of the analysis and the reliability of the study conclusions.

This study was conducted in accordance with the Declaration of Helsinki (1964) and was approved by the Medical Ethics Committee of Union Hospital, Tongji Medical College, Huazhong University of Science and Technology, with a waiver of informed consent. The permit number is: [2024] Lun Shen Zi (0626).

## 2.2. Data collection

Medical data were collected from all participants, including age, gender, height, weight, smoking history, alcohol consumption history, admission blood pressure, and medical history (including coronary heart disease history, hypertension history, autoimmune disease history, thyroid disease history, tumor history, liver and kidney dysfunction history, etc.), as well as the duration of diabetes and medication history. Body mass index (BMI) was calculated as weight (kg)/ height (m)^2.

After an overnight fast of more than 8 hours, venous blood samples were collected from each participant to measure neutrophils, platelets, lymphocytes, monocytes, FBG, HbA1c, total cholesterol (TC), triglycerides (TG), low-density lipoprotein cholesterol (LDL-C), high-density lipoprotein cholesterol (HDL-C), alanine transaminase (ALT); aspertate aminotransferase (AST), ALB, etc. The UAR, Monocyte to HDL Ratio (MHR) and triglyceride glucose index (TyG index) were calculated using the following formulas: UAR = Uric acid (μmol/L)/ Albumin (g/L); MHR = Monocyte ($10^9$/L) / HDL (mmol/L); TyG index = Ln [Triglyceride (TG, mg/dl) × Fasting plasma glucose (mg/dl)/2].

## 2.3. Carotid atherosclerosis assessment

Carotid ultrasound examinations were conducted using a color Doppler ultrasound diagnostic instrument by trained and certified ultrasound specialists. Imaging was performed using the same color Doppler device equipped with a 7.0–10 MHz transducer. All patients were placed in a supine position, with their heads slightly turned toward the opposite side of the examination and tilted backward to fully expose the carotid arteries.

Bilateral assessments of the common carotid artery (CCA) were performed both transversely and longitudinally, including the proximal CCA, distal CCA, bulb, internal carotid artery (ICA), and external carotid artery (ECA). Scanning was conducted in various directions to evaluate atherosclerotic plaques and stenosis, as well as to measure intima-media thickness

(IMT). The carotid intima-media thickness (CIMT) was measured between the intimal-luminal and medial-adventitial interfaces of the carotid artery wall. The average CIMT was calculated as the mean of the IMT values from both carotid arteries, while the maximum CIMT was defined as the larger IMT value between the two sides. Abnormal CIMT was defined as either the average or maximum CIMT value ≥ 1 mm [21,22].

In this study, abnormal CIMT (CIMTA) was specifically defined as a maximum CIMT value ≥ 1 mm. Carotid artery plaques (CAP) were defined as focal structures invading the arterial lumen, focal thickening of the arterial wall exceeding 50%, or a CIMT > 1.5 mm [23]. Carotid atherosclerosis (CAS) encompasses both CIMTA and CAP [9].

### 2.4. Patient grouping

1. Patients were divided into two groups based on ultrasound findings: the CAS group and the WCAS group.

2. Given that carotid intima-media thickness (CIMT) and carotid plaques (CAP) can predict the risk of cardiovascular disease (CVD) [24], the CAS group was further divided into the CIMTA group and the CAP group.

3. The CAP group was subdivided into single carotid plaque group (SC group) and multiple or bilateral plaque group (MC group) based on plaque characteristics [25].

### 2.5. Statistical analysis

Statistical analysis of the collected data was performed using SPSS version 26.0, and graphs were generated using Origin software. The normality of continuous variables was assessed using the Shapiro-Wilk test. Normally distributed continuous variables were presented as mean ± standard deviations (SDs) and compared between groups using t-tests (for two groups) or one-way ANOVA (for three groups). Non-normally distributed continuous variables were described as median (interquartile range) and compared using the Mann-Whitney U test (for two groups) or Kruskal-Wallis H test (for three groups). Categorical variables were expressed as numbers and percentages, with chi-square tests used to evaluate statistical differences among groups. Spearman correlation analysis was employed to examine the relationship between UAR and other clinical variables in CAS patients. Both univariate and multivariate logistic regression analyses were conducted to explore the association between UAR and CAS. Receiver operating characteristic (ROC) curve analysis was used to determine the optimal cutoff value for UAR in identifying CAS risk, defined by maximizing the Youden index. A two-tailed P value < 0.05 was considered statistically significant.

## 3. Result

### 3.1. Baseline characteristics of patients in the CAS and WCAS groups

The demographic and clinical characteristics of patients in the T2DM-CAS and T2DM-WCAS groups are summarized in Table 1. Among the 390 enrolled diabetic patients, 259 had carotid atherosclerosis (CAS) and 131 did not (WCAS). Compared to the WCAS group, the CAS group had a higher prevalence of hyperlipidemia and hypertension. Significant statistical differences were observed between the two groups in terms of age, duration of diabetes, and smoking history (P < 0.05). Additionally, compared to the WCAS group, the CAS group exhibited significantly higher levels of systolic blood pressure (SBP), UA, creatinine (Cr), blood urea nitrogen (BUN), and UAR (P < 0.05). Conversely, the estimated glomerular

**Table 1. Demographic and clinical data of diabetic subjects with and without CAS.**

| Variables | WCAS (n = 131) | CAS (n = 259) | P value |
|---|---|---|---|
| Gender (male, %) | 87(66.4%) | 190(73.4%) | 0.153 |
| Age (years) | 45.00 ± 11.81 | 54.28 ± 10.81 | **<0.001** |
| BMI (kg/m2) | 25.13 ± 3.34 | 25.29 ± 3.22 | 0.654 |
| Diabetes duration (years) | 4(1,8) | 5(1,10) | **0.015** |
| Smoking, n (%) | 38(29.2%) | 107(41.3%) | **0.020** |
| Alcohol, n (%) | 27(20.9%) | 61(23.6%) | 0.561 |
| Hypertension, n (%) | 46(35.4%) | 119(45.9%) | **0.047** |
| SBP (mmHg) | 126.79 ± 15.86 | 131.13 ± 15.70 | **0.011** |
| DBP (mmHg) | 81.42 ± 10.04 | 79.99 ± 10.20 | 0.192 |
| Monocyte, 109/L | 0.40(0.30,0.49) | 0.41(0.34,0.50) | 0.065 |
| FBG (mmol/L) | 7.5(6.3,9.8) | 7.4(6.3,9.4) | 0.656 |
| HbA1c (%) | 9.00(7.50,10.90) | 8.70(7.30,10.35) | 0.246 |
| TG, mmol/L | 1.47(0.95,2.90) | 1.60(1.03,2.47) | 0.937 |
| TC, mmol/L | 4.80(4.19,5.50) | 4.85 ± 1.21 | 0.762 |
| HDL-C, mmol/L | 1.13 ± 0.32 | 1.00(0.85,1.18) | **0.012** |
| LDL-C, mmol/L | 2.74 ± 0.95 | 2.81 ± 0.98 | 0.386 |
| ALB (g/L) | 41.30(39.4,44.8) | 40.51 ± 3.39 | **<0.001** |
| ALT (U/L) | 22.5(16,41) | 21(15,34) | 0.106 |
| AST (U/L) | 19(15.75,25.25) | 19.5(16,25) | 0.677 |
| UA(μmol/L) | 297.81 ± 66.00 | 333.4(290.2,394.9) | **<0.001** |
| BUN (mmol/L) | 5.08(4.35,6.14) | 5.45(4.64,6.68) | **0.024** |
| Cr(μmol/L) | 64.46 ± 14.57 | 67.15(58.08,78.23) | 0.052 |
| eGFR (mL/m/1.73 m2) | 108.14(99.63,118.33) | 99.83(89.54,110.21) | **<0.001** |
| Hyperlipidemia, n (%) | 86(66.2%) | 196(75.7%) | **0.047** |
| Fatty liver, n (%) | 82(66.7%) | 169(71.6%) | 0.332 |
| TyG index | 1.94 ± 0.95 | 1.81(1.29,2.30) | 0.669 |
| UAR | 7.11 ± 1.49 | 8.29(7.01,9.64) | **<0.001** |
| CAS | | | |
| CIMTA, n (%) | / | / | 91(35.1%) |
| CAP, n (%) | / | / | 168(64.9%) |

SBP, systolic blood pressure; DBP, diastolic blood pressure; FBG, fasting blood glucose; HbA1c, glycosylated hemoglobin; TG, triglyceride; TC, total cholesterol; HDL-C, high-density lipoprotein cholesterol; LDL-C, low-density lipoprotein cholesterol; ALB, albumin; ALT, alanine transaminase; AST, aspertate aminotransferase; UA, uric acid; BUN, blood urea nitrogen; eGFR, estimated glomerular filtration; TyG index, triglyceride glucose index; UAR, UA/ALB ratio. $P < 0.05$ (two-sided) was defined as statistically significant. Bold values indicate statistically significance.

filtration (eGFR), HDL-C, and ALB levels were significantly lower in the CAS group (P < 0.05). No significant differences were found between the two groups regarding gender, BMI, alcohol consumption history, diastolic blood pressure (DBP), FBG, HbA1c, TG, TC, LDL-C, ALT, AST, and the TyG index (P > 0.05). In the CAS group, the CIMTA and CAP groups accounted for 35.1% and 64.9%, respectively.

The box plot in Fig 1 shows that UAR in the CAS group was significantly higher than that in the WCAS group (P < 0.001).On the box plots, central lines represent the median, the length of the box represents the interquartile range and the lines extend to minimum and maximum values. P < 0.05 (two-sided) was defined as statistically significant.

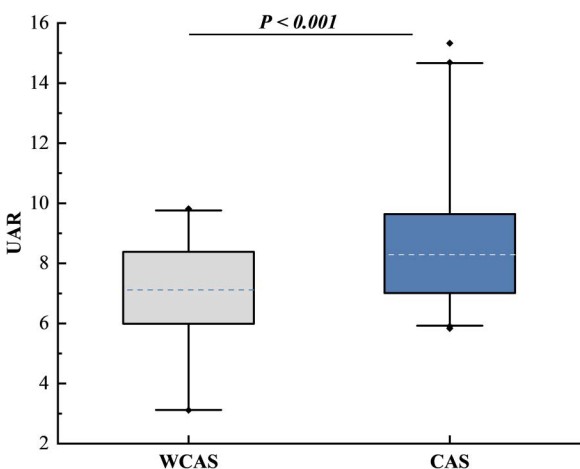

**Fig 1. Distribution of UAR between the WCAS and CAS group.**

### 3.2. Baseline characteristics of patients in the WCAS, CIMTA, and CAP groups

As shown in Table 2, there were statistically significant differences among the three groups regarding age, duration of diabetes, prevalence of hypertension, SBP, HDL-C, ALB, ALT, AST, UA, BUN, eGFR, and UAR levels (all P < 0.05). However, no significant differences were found in gender, BMI, alcohol consumption history, DBP, FBG, HbA1c, TG, TC, LDL-C, Cr, TyG index, or the prevalence of hyperlipidemia and fatty liver (all P > 0.05).

The box plots presented in Fig 2 illustrate the distribution of the UAR across the WCAS, CIMTA, and CAP groups. The results indicate that both the CIMTA and CAP groups exhibit statistically significant differences when compared to the WCAS group (both P < 0.05). However, no statistically significant difference was observed between the CIMTA and CAP groups (P > 0.05).

The box plot in the Fig 2 displays the median as the central line, the interquartile range as the length of the box, and the lines extending to the minimum and maximum values. * P < 0.05 versus WCAS; # P > 0.05 versus CAP. *P* < 0.05 (two-sided) was defined as statistically significant.

### 3.3. Clinical and laboratory characteristics of T2DM-CAS patients: subgroup analysis based on CAS plaque characteristics

As presented in Table 3, patients in the MC group had significantly higher age, prevalence of hypertension, monocyte count, HDL-C, and Cr levels compared to those in the SC group (P < 0.05). Conversely, the MC group showed significantly lower DBP, HbA1c, and eGFR levels than the SC group (P < 0.05). The P value for MHR between the two groups was 0.051, approaching statistical significance. No significant differences were found between the groups in terms of gender, BMI, duration of diabetes, smoking history, alcohol consumption history, SBP, FBG, HbA1c, TG, TC, HDL-C, LDL-C, ALB, ALT, AST, UA, BUN, TyG index, UAR, or prevalence of hyperlipidemia and fatty liver (all P > 0.05).

### 3.4. Correlation of UAR with other variables in the CAS group

Spearman correlation analysis was performed to assess the relationship between UAR and other indicators in T2DM-CAS patients. (Table 4). UAR exhibited statistically significant

**Table 2. Demographic and clinical data of WCAS、CIMTA and CAP groups.**

| Variables | WCAS (n = 131) | CIMTA (n = 91) | CAP (n = 168) | P value |
|---|---|---|---|---|
| Gender (male, %) | 87(66.4%) | 70(76.9%) | 120(71.4%) | 0.234 |
| Age(years) | 45 ± 11.81 | 49.73 ± 10.78 | 57(51, 64) | **<0.001** |
| BMI (kg/m2) | 25.13 ± 3.34 | 25.69 ± 3.40 | 25.07 ± 3.12 | 0.310 |
| Diabetes duration (years) | 4(1, 8) | 4(1, 9) | 6(1, 11) | **0.007** |
| Smoking, n (%) | 38(29.2%) | 41(45.1%) | 66(39.3%) | **0.044** |
| Alcohol, n (%) | 27(20.9%) | 25(27.5%) | 36(21.4%) | 0.457 |
| Hypertension, n (%) | 46(35.4%) | 32(35.2%) | 87(51.8%) | **0.005** |
| SBP (mmHg) | 126.79 ± 15.86 | 129.55 ± 15.20 | 131.99 ± 15.94 | **0.019** |
| DBP (mmHg) | 81.42 ± 10.04 | 81.69 ± 9.7 | 79.07 ± 10.34 | 0.059 |
| FBG (mmol/L) | 7.5(6.3, 9.8) | 7.25(6.1, 9.1) | 7.6(6.4, 9.5) | 0.421 |
| HbA1c (%) | 9(7.5, 10.9) | 9(7.08, 10.73) | 8.4(7.4, 10.2) | 0.461 |
| TG, mmol/L | 1.47(0.95, 2.90) | 1.76(1.19, 2.79) | 1.48(1.02, 2.24) | 0.278 |
| TC, mmol/L | 4.80(4.19, 5.50) | 4.82(4.11, 5.60) | 2.31 ± 2.89 | 0.751 |
| HDL-C, mmol/L | 1.09(0.88, 1.34) | 0.98(0.82, 1.16) | 1.01(0.86, 1.19) | **0.027** |
| LDL-C, mmol/L | 2.74 ± 0.95 | 2.84 ± 0.92 | 2.80 ± 1.01 | 0.735 |
| ALB (g/L) | 41.92 ± 3.77 | 40.88 ± 3.93 | 39.9 (38.13, 42.28) | **0.001** |
| ALT (U/L) | 22.5(16, 41) | 24(16, 40.75) | 20.5(14.25, 31) | **0.015** |
| AST (U/L) | 19(15.75, 25.25) | 21(17, 27) | 18(15.25, 25.00) | **0.019** |
| UA (μmol/L) | 296.4(248.1, 348.1) | 351.10 ± 74.09 | 327.9(279.85, 397.68) | **<0.001** |
| BUN (mmol/L) | 5.08(4.35, 6.14) | 5.46(4.33, 6.60) | 5.74 ± 1.59 | **0.050** |
| Cr (μmol/L) | 65.5(54.18, 75.15) | 67.61 ± 13.55 | 69.37 ± 16.75 | 0.123 |
| eGFR (mL/m/1.73m2) | 108.14(99.63,118.33) | 104.08(94.64,115.02) | 96.68 ± 15.76 | **<0.001** |
| Hyperlipidemia, n (%) | 86(66.2%) | 69(75.8%) | 127(75.6%) | 0.140 |
| Fatty liver, n (%) | 82(66.7%) | 60(72.3%) | 109(71.2%) | 0.616 |
| TyG index | 1.78(1.26, 2.61) | 1.91 ± 0.80 | 1.85 ± 0.83 | 0.640 |
| UAR | 7.12(5.99, 8.38) | 8.32(7.31, 9.83) | 8.22(6.94, 9.57) | **<0.001** |

SBP, systolic blood pressure; DBP, diastolic blood pressure; FBG, fasting blood glucose; HbA1c, glycosylated hemoglobin; TG, triglyceride; TC, total cholesterol; HDL-C, high-density lipoprotein cholesterol; LDL-C, low-density lipoprotein cholesterol; ALB, albumin; ALT, alanine transaminase; AST, aspertate aminotransferase; UA, uric acid; BUN, blood urea nitrogen; eGFR, estimated glomerular filtration; TyG index, triglyceride glucose index; UAR, UA/ALB ratio. $P < 0.05$ (two-sided) was defined as statistically significant. Bold values indicate statistically significance.

positive correlations with gender(r = 0.133, P < 0.05), BMI (r = 0.176, P < 0.05), smoking (r = 0.102, P < 0.05), hypertension (r = 0.162, P < 0.05), TG (r = 0.214, P < 0.05), UA (r = 0.917, P < 0.001), BUN (r = 0.123, P < 0.05), Cr (r = 0.245, P < 0.05), TyG index (r = 0.186, P < 0.05), and prevalence of fatty liver (r = 0.164, P < 0.05). In contrast, UAR showed significant negative correlations with HDL-C (r = -0.256, P < 0.001), ALB (r = -0.199, P < 0.001), and eGFR (r = -0.182, P < 0.001). No significant correlations were found between UAR and age, duration of diabetes, alcohol consumption, SBP, DBP, FBG, HbA1c, TC, LDL-C, ALT, AST, or prevalence of hyperlipidemia (all P > 0.05).

### 3.5. Univariate and multivariate logistic regression analysis of UAR for T2DM-CAS occurrence

Univariate logistic regression analysis revealed that age, duration of diabetes, smoking history, hypertension, SBP, Cr, eGFR, UAR, and hyperlipidemia (HLP) were independently associated with the incidence of CAS in T2DM patients (Table 5). After controlling for confounding

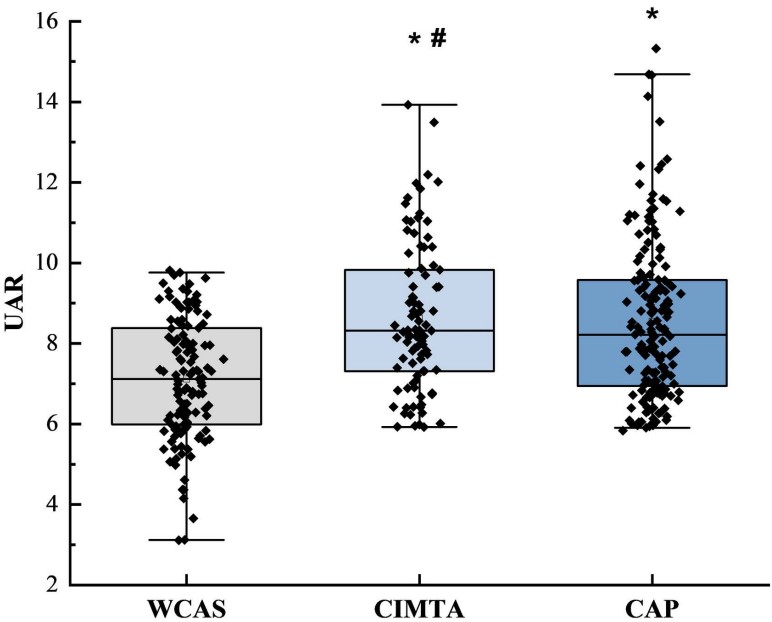

**Fig 2. Distribution of UAR Across Different Groups.**

factors, age, smoking history, SBP, HLP, and UAR remained statistically significant and were identified as independent risk factors for the occurrence of CAS in T2DM patients (all P < 0.05).

### 3.6  Diagnostic capability of UAR for T2DM-CAS

ROC curve analysis was employed to evaluate the ability of UAR to identify T2DM-CAS patients. Fig 4 indicates that UAR demonstrated a high predictive value for T2DM-CAS (AUC = 0.712). The optimal UAR cutoff value for predicting CAS occurrence was found to be 7.62. Additionally, the AUC for the combination model of UAR and age was 0.819, while the AUC for the combination model of UAR with age, smoking, SBP, and hyperlipidemia was 0.829.

### 3.7.  Distribution of UAR based on the optimal cutoff value

To assess the validity of the cutoff point obtained from the ROC curve analysis, we further analyzed the distribution of UAR between the two groups. Based on the cutoff value of 7.62, the percentage of patients with low UAR (L-UAR) in the WCAS group was higher than that of high UAR (H-UAR), whereas the CAS group exhibited a higher percentage of H-UAR compared to L-UAR (Fig 5A). We also compared the proportional distribution of disease status between L-UAR and H-UAR patients. Among the H-UAR subjects, nearly 80% were from the CAS group, while there was little difference between the WCAS and CAS groups in the L-UAR group (Fig 5B).

### 4.  Discussion

Currently, there is no evidence linking UAR to T2DM-CAS. This retrospective study primarily explores the association between UAR and patients with T2DM-CAS. Our findings indicate that elevated UAR levels are closely related to the occurrence of carotid atherosclerosis in T2DM patients and serve as an independent predictor for T2DM-CAS. ROC curve analysis

**Table 3. Characteristics of subjects in different carotid plaque groups.**

| Variables | SC group (n = 55) | MC group (n = 113) | P value |
|---|---|---|---|
| Gender (male, %) | 38 (69.1%) | 82 (72.6%) | 0.640 |
| Age(years) | 53(43, 60) | 58.81 ± 9.22 | **0.001** |
| BMI (kg/m2) | 25.06 ± 3.14 | 25.08 ± 3.13 | 0.983 |
| Diabetes duration (years) | 6 (1, 10) | 6 (1, 12) | 0.960 |
| Smoking, n (%) | 20(36.4%) | 46(40.7%) | 0.588 |
| Alcohol, n (%) | 12(21.8%) | 24(21.2%) | 0.932 |
| Hypertension, n (%) | 22(40.0%) | 65(57.5%) | **0.033** |
| SBP (mmHg) | 133.42 ± 15.71 | 131.29 ± 16.07 | 0.419 |
| DBP (mmHg) | 82.07 ± 9.62 | 77.60 ± 10.40 | **0.008** |
| Monocyte, 109/L | 0.41 (0.30, 0.46) | 0.45 ± 0.13 | **0.023** |
| FBG (mmol/L) | 7.7 (6.15, 10.15) | 7.6 (6.5, 9.4) | 0.805 |
| HbA1c (%) | 9.2 (7.83, 10.9) | 8.3 (7.3, 9.9) | **0.020** |
| TG, mmol/L | 1.56 (1.04, 2.57) | 1.45 (0.98, 2.09) | 0.340 |
| TC, mmol/L | 5.02 ± 1.35 | 4.61 (3.89, 5.42) | 0.170 |
| HDL-C, mmol/L | 0.96 (0.86, 1.17) | 1.04 (0.87, 1.2) | 0.463 |
| LDL-C, mmol/L | 2.81 ± 0.98 | 2.79 ± 1.02 | 0.888 |
| ALB (g/L) | 40.56 ± 3.27 | 40.19 ± 2.96 | 0.469 |
| ALT (U/L) | 22 (17, 29) | 19 (14, 33.50) | 0.240 |
| AST (U/L) | 19 (16, 25) | 18 (15, 25) | 0.419 |
| UA (μmol/L) | 342.62 ± 74.37 | 325.80 (277.70, 399.70) | 0.862 |
| BUN (mmol/L) | 5.43 ± 1.27 | 5.5 (4.76, 6.88) | 0.297 |
| Cr (μmol/L) | 65.28 ± 13.67 | 69.3 (58.35, 82.4) | **0.041** |
| eGFR (mL/m/1.73 m2) | 102.76 ± 13.50 | 93.46 (85.54, 105.11) | **<0.001** |
| Hyperlipidemia, n (%) | 41(74.5%) | 86(76.1%) | 0.825 |
| Fatty liver, n (%) | 39(76.5%) | 70(68.6%) | 0.312 |
| TyG index | 1.95 ± 0.92 | 1.76 (1.35, 2.11) | 0.452 |
| MHR, 109/mmol/L | 0.39 ± 0.14 | 0.43 (0.33, 0.53) | 0.051 |
| UAR | 8.21 (7.26, 9.31) | 8.28 (6.81, 9.61) | 0.915 |

TyG index, triglyceride glucose index; MHR, Monocyte/HDL-C ratio; UAR, UA/ALB ratio. SC group, single carotid plaque; MC group, bilateral or multiple. $P < 0.05$ (two-sided) was defined as statistically significant. Bold values indicate statistically significance.

shows that UAR can effectively predict the occurrence of T2DM-CAS. Furthermore, combining UAR with age, smoking, SBP, and hyperlipidemia enhances predictive capability, achieving an AUC of 0.829. This suggests that a comprehensive model incorporating UAR and other clinical variables may be more effective.

Atherosclerosis is a chronic inflammatory disease characterized by lipid accumulation and inflammation in arterial walls [26]. It accounts for over 80% of deaths and disabilities in T2DM patients, making it a major contributor to macrovascular complications [27]. At its advanced stage, atherosclerosis manifests as lesions in the intima and the accumulation of plaques [7]. Longitudinal studies have demonstrated that CIMT and CAP are independent predictors of cardiovascular events [28,29]. This study investigates the relationship between UAR and CAS in T2DM patients, further categorizing CAS patients into abnormal CIMTA and CAP groups for analysis. The goal is to identify easily accessible blood biomarkers for early screening of T2DM patients to prevent further vascular complications.

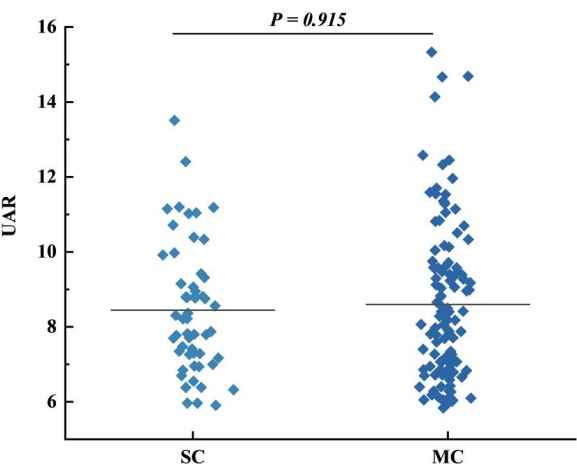

**Fig 3. Distribution of UAR between the SC and MC group.** The Fig 3 illustrates the differences in UAR between the SC and MC groups. In the scatter plot, the data were presented as median (min -max). $P < 0.05$ (two-sided) was defined as statistically significant.

CAS is the most common form of large vessel disease and a leading cause of ischemic stroke [30]. Evidence suggests that UA contributes to the atherosclerotic process by disrupting lipid metabolism, reducing nitric oxide synthesis in endothelial cells, promoting vascular smooth muscle cell proliferation, and enhancing inflammation [31]. ALB, synthesized by hepatocytes, is influenced by inflammation and nutritional status, acting as a negative acute phase protein correlated inversely with the severity of inflammation, low levels of ALB are associated with atherosclerosis, as it maintains osmotic pressure, resists blood stagnation and thrombosis, and provides antioxidant effects [32,33]. Previous studies have also shown that low serum ALB is associated with cardiovascular disease, which results in loss of antioxidant capacity as well as anticoagulant and antiplatelet activity [19]. The UAR has been identified as a prognostic factor for adverse cardiovascular events, serving as a new marker for assessing inflammation and oxidative stress [34]. Notably, UAR is inexpensive and easily obtainable. Liu et al. found that UAR was associated with in-stent restenosis after coronary artery stent implantation [35]. Wang et al. showed that UAR is a predictor of prognosis in aortic dissection [36]. Li et al. found that UAR can predict long-term cardiac mortality in patients with unstable angina after percutaneous coronary intervention [37]. These findings collectively indicate a significant relationship between UAR and cardiovascular diseases, validating its diagnostic potential across various conditions.

In our study, box plots reveal significant differences in UAR levels between the WCAS and CAS groups. The elevated levels of UAR, smoking history, hypertension, and SBP in the CAS group suggest that higher UAR may correlate with more severe vascular lesions. After further categorizing CAS into CAP and CIMTA groups, we found that higher UAR levels were independently associated with CAP and CIMTA in T2DM patients, respectively. Notably, there was no statistical difference in UAR levels between the CIMTA and CAP groups, indicating that UAR may have predictive value even in the early stages of carotid intima-media change. Previous studies have also shown that inhibiting the progression of CIMT can reduce the occurrence of cardiovascular events [38]. The analysis of plaque characteristics revealed significant differences in monocyte counts between the groups. Monocytes promote atherosclerosis progression through the release of pro-inflammatory cytokines, reactive oxygen species, and proteolytic enzymes [39]. Although the difference in MHR between SC and MC groups did

**Table 4. Correlation of the UAR with other parameters in the T2DM-CAS patients.**

| UAR | | |
|---|---|---|
| | r | P |
| Gender (male, %) | 0.133 | **0.008** |
| Age (years) | 0.067 | 0.186 |
| BMI (kg/m2) | 0.176 | **<0.001** |
| Diabetes duration (years) | - 0.066 | 0.191 |
| Smoking, n (%) | 0.102 | **0.043** |
| Alcohol, n (%) | 0.076 | 0.137 |
| Hypertension, n (%) | 0.162 | **0.001** |
| SBP (mmHg) | 0.002 | 0.964 |
| DBP (mmHg) | 0.029 | 0.569 |
| FBG (mmol/L) | -0.032 | 0.540 |
| HbA1c (%) | -0.072 | 0.174 |
| TG, mmol/L | 0.214 | **<0.001** |
| TC, mmol/L | 0.042 | 0.412 |
| HDL-C, mmol/L | -0.256 | **<0.001** |
| LDL-C, mmol/L | 0.040 | 0.428 |
| ALB (g/L) | -0.199 | **<0.001** |
| ALT (U/L) | 0.020 | 0.700 |
| AST (U/L) | 0.067 | 0.190 |
| UA (μmol/L) | 0.917 | **<0.001** |
| BUN (mmol/L) | 0.123 | **0.015** |
| Cr (μmol/L) | 0.245 | **<0.001** |
| eGFR (mL/m/1.73 m2) | -0.182 | **<0.001** |
| HLP, n (%) | 0.094 | 0.063 |
| Fatty liver, n (%) | 0.164 | **0.002** |
| TyG index | 0.186 | **0.001** |

SBP, systolic blood pressure; DBP, diastolic blood pressure; FBG, fasting blood glucose; HbA1c, glycosylated hemo-globin; TG, triglyceride; TC, total cholesterol; HDL-C, high-density lipoprotein cholesterol; LDL-C, low-density lipoprotein cholesterol; ALT, alanine transaminase; AST, aspertate aminotransferase; ALB, albumin; UA, uric acid; Cr, creatinine;BUN, blood urea nitrogen; eGFR, estimated glomerular filtration; TyG index, triglyceride glucose index; UAR, UA/ALB ratio. $P < 0.05$ (two-sided) was defined as statistically significant. Bold values indicate statistically significance.

**Table 5. Univariate and binary logistic regression analysis results.**

| | Variable OR (95% CI) | P value | Variable OR (95%CI) | P value |
|---|---|---|---|---|
| Age | 1.074(1.052~1.097) | **<0.001** | 1.153(1.089~1.221) | **<0.001** |
| Diabetes duration | 1.058(1.017~1.100) | **0.005** | | |
| Smoking | 1.704(1.085~2.677) | **0.021** | 1.856(1.013~3.400) | **0.045** |
| Hypertension | 1.552(1.005~2.397) | **0.047** | | |
| SBP | 1.018(1.004~1.032) | **0.011** | 1.024(1.004~1.044) | **0.017** |
| Cr | 1.019(1.004~1.034) | **0.010** | | |
| eGFR | 0.969(0.956~0.983) | **<0.001** | | |
| HLP | 1.592(1.004~2.524) | **0.048** | 1.880(1.034~3.421) | **0.039** |
| UAR | 1.691(1.448~1.975) | **<0.001** | 1.849(1.525~2.243) | **<0.001** |

SBP, systolic blood pressure; ALB, albumin; Cr, creatinine; eGFR, estimated glomerular filtration; TyG index, triglyceride glucose index; UAR, UA/ALB ratio. $P < 0.05$ (two-sided) was defined as statistically significant. Bold values indicate statistically significance.

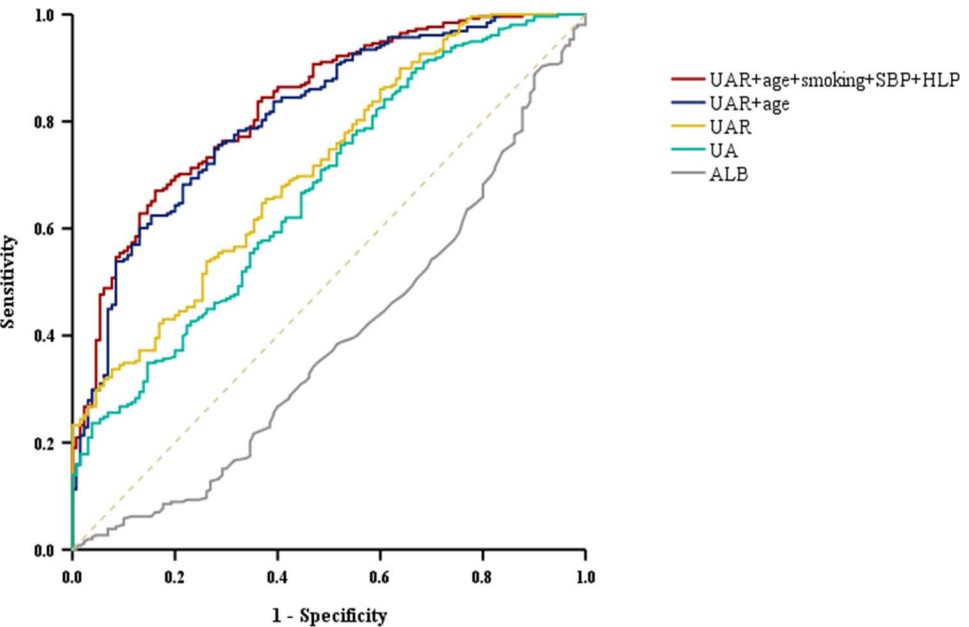

**Fig 4. ROC Curve Analysis for Predicting T2DM-CAS.** UAR (AUC 0.712, 95% CI:0.660–0.765, *P* = 0.000, cut-off 7.62, sensitivity 65.5%, specificity 62.3%); UAR+age+smoking+SBP+HLP (AUC 0.829, 95% CI:0.787–0.871, *P* = 0.000, cut-off 0.73, sensitivity 67%, specificity 90.8%); UAR+age (AUC 0.819, 95% CI:0.776–0.863, *P* = 0.000, cut-off 0.74, sensitivity 65.1%, specificity 85.4%); UA (AUC 0.671, 95% CI:0.615–0.727, *P* = 0.000, cut-off 289.7, sensitivity 75.6%, specificity 47.7%); ALB (AUC 0.389, 95% CI:0.329–0.449, *P* = 0.000, cut-off 43.5, sensitivity 17.4%, specificity 65.4%).

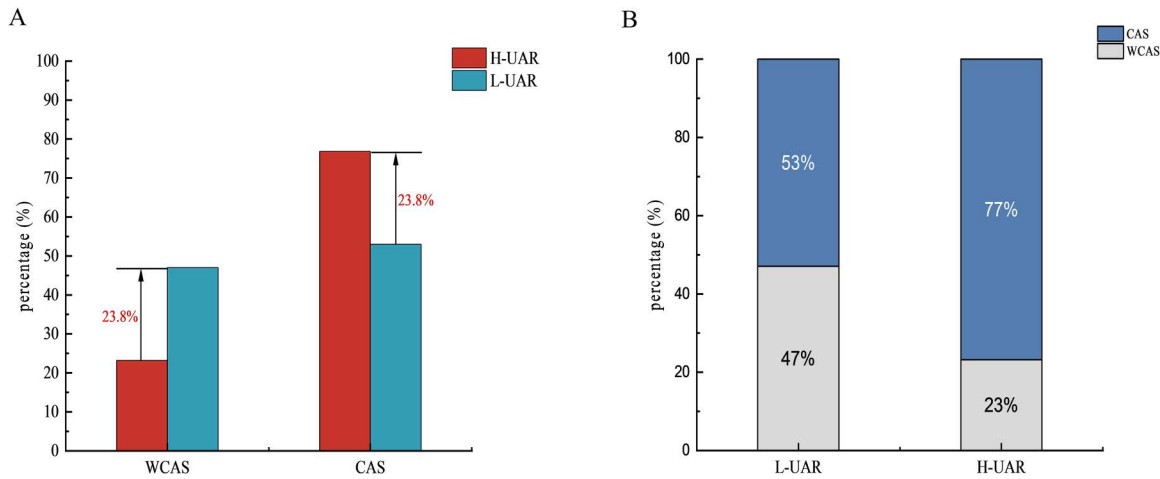

**Fig 5. UAR Distribution Based on the Cut-off Point Determined by ROC Curve Analysis.** (A) The proportions of low and high UAR in the WCAS and CAS groups are displayed, highlighting the distinct distribution between these groups. (B) The proportion distribution of the two groups in patients with low and high UAR.

not reach statistical significance, its proximity suggests it may play a role in assessing diabetes-related carotid atherosclerosis, aligning with prior studies [40].

Spearman analysis demonstrated a significant correlation between UAR and renal function decline in T2DM patients (elevated BUN and Cr levels, decreased eGFR). Other studies have indicated that UAR can predict renal failure in Chinese IgA nephropathy patients [41]. Additionally,

A population-based study showed that CIMT and CAP were associated with decreased renal function and CKD [42]. Studies have shown that TG is associated with endothelial dysfunction and subclinical atherosclerosis [43], both SUA and TG are associated with the presence of carotid plaque, and their combination increases the risk of carotid plaque [44]. HDL-C has the effect of reverse cholesterol transport, which can reduce atherosclerosis, anti-thrombosis, anti-inflammation, vasodilation and anti-apoptosis [45]. In this study, there was also a significant positive correlation between UAR and TG and SUA and a significant negative correlation between UAR and HDL-C levels. We also observed a significant positive correlation between UAR and both gender and BMI. Through gender-stratified analysis (see S1 Fig), the results indicated that UAR was significantly associated with outcomes in both male and female patients. However, after adjusting for potential confounders, the risk increase was more pronounced in females. This finding highlights UAR as an important risk factor and suggests a significant gender-based difference in its impact. Obesity is a metabolic disease that induces insulin resistance by altering the insulin signaling pathway through a mild inflammatory state [46]. UAR may promote atherosclerosis in patients by exacerbating insulin resistance, as evidenced by its correlation with the TyG index, a surrogate marker for insulin resistance [47]. A recent study also showed that UAR is associated with obesity and insulin resistance in adolescents [48]. Despite hyperglycemia enhancing vascular inflammation [49], our data did not show a significant correlation between UAR and fasting glucose levels in T2DM patients, possibly due to sample size limitations.

Univariate logistic regression analysis revealed that age, duration of diabetes, smoking history, hypertension, SBP, Cr, eGFR, HLP, and UAR were statistically significant factors. Multivariate regression confirmed that age, smoking history, SBP, HLP, and UAR were independently associated with T2DM-CAS. This study also showed that age and smoking history may be independent risk factors, which is consistent with previous studies [50]. After performing a stratified analysis of age in this study (see S2 Fig), it was found that the association between UAR and T2DM-CAS was significant across different age groups. Among all age groups, the 60-80 age group showed a significant increase in odds ratio after controlling for potential confounding factors. This suggests that special attention should be given to treatment strategies for the elderly population in clinical applications. ROC curve analysis demonstrated that UAR effectively predicts T2DM-CAS (AUC = 0.712), surpassing the predictive ability of UA (AUC = 0.671) or ALB (AUC = 0.389) alone. Additionally, the combination of UAR with other independent risk factors showed higher predictive efficiency, with an AUC of 0.819 when combined with age and 0.829 when combined with age, smoking, SBP, and hyperlipidemia. Clinicians can leverage this comprehensive approach to more accurately identify high-risk patients and implement targeted interventions to improve outcomes. The distribution analysis of L-UAR and H-UAR patients revealed that nearly 80% of H-UAR subjects were in the CAS group, while the proportions of WCAS and CAS groups were similar in the L-UAR group. This suggests that L-UAR may not serve as a protective factor against CAS in T2DM patients, while excessively high UAR levels could promote the development of carotid atherosclerosis through certain mechanisms.

## 5.  Limitations

This study also has several limitations. First, it is a retrospective cross-sectional study conducted at a single center, which prevents us from establishing a causal relationship between UAR and T2DM-CAS. Secondly, our sample size is relatively small, and we only investigated this relationship in T2DM subjects, which may lead to less precise estimates and limit the generalizability of our findings. Therefore, the results need to be validated in other types of diabetes and in healthy subjects. Additionally, in this study, we could not completely rule out other potential confounding factors. For example, we did not conduct stratified analyses based

on the use of hypoglycemic agents, education level, social status, and economic level between the two groups of patients, nor did we incorporate psychological factors into the analysis. Third, the hematological parameters used in the study are static and only represent the status at a specific point in time, limiting their ability to reflect long-term trends. Future research should consider a large-scale multicenter prospective design, conduct more detailed stratified analyses, and incorporate dynamic monitoring data to validate the long-term efficacy and general applicability of UAR as a predictive biomarker. Additionally, exploring the role of UAR in other diabetes-related complications could enhance our understanding of its potential mechanisms.

## 6. Conclusion

In summary, the results indicate that UAR levels are significantly elevated in T2DM patients with CAS compared to those without this complication. Even after adjusting for relevant confounding factors, a significant association between UAR and CAS persists. ROC curve analysis demonstrates that UAR is a good predictor of T2DM-CAS (AUC = 0.712). Combining UAR with other clinical parameters, such as age, smoking history, SBP, and HLP may assist clinicians in more accurately identifying high-risk patients, enabling targeted interventions to improve patient outcomes. However, further research is needed to confirm the predictive value of UAR in T2DM-CAS.

## Supporting information

**S1 Dataset. The data set used to produce the study's findings.**
(SAV)

**S1 Fig. Association between UAR and T2DM-CAS in overall, female and male.**
(TIF)

**S2 Fig. Association between UAR and T2DM-CAS at different age stages.**
(TIF)

## Author contributions

**Data curation:** Yao Yin, Liyin Zhang.

**Writing – original draft:** Yao Yin.

**Writing – review & editing:** Si Jin, Jiaoyue Zhang.

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
