## [Decision Letter · Decision Letter 0]

22 Jan 2025

PONE-D-24-50053Predictive Value of Uric Acid to Albumin Ratio for Carotid Atherosclerosis in Type 2 Diabetes Mellitus: A retrospective studyPredictive Value of Uric Acid to Albumin Ratio for Carotid Atherosclerosis in Type 2 Diabetes MellitusPLOS ONE

Dear Dr. Jin,

Thank you for submitting your manuscript to PLOS ONE. After careful consideration, we feel that it has merit but does not fully meet PLOS ONE’s publication criteria as it currently stands. Therefore, we invite you to submit a revised version of the manuscript that addresses the points raised during the review process.

We look forward to receiving your revised manuscript.

Kind regards,

Aleksandra Klisic

Academic Editor

PLOS ONE

“This work was supported by the National Natural Science Foundation of China (Grant No. 82070862, No.82370840) and the National Key R&D Program of China (Grant No. 2020YFC2008900). These funds were obtained by Professor Jin.”

3. In the online submission form, you indicated that [The datasets used and analyzed during the current study are available from the corresponding author upon reasonable request.].

Reviewers' comments:

Reviewer's Responses to Questions

**Comments to the Author**

1. Is the manuscript technically sound, and do the data support the conclusions?

Reviewer #1: Yes

Reviewer #2: Yes

Reviewer #3: Yes

2. Has the statistical analysis been performed appropriately and rigorously? 

Reviewer #1: Yes

Reviewer #2: Yes

Reviewer #3: Yes

3. Have the authors made all data underlying the findings in their manuscript fully available?

Reviewer #1: Yes

Reviewer #2: No

Reviewer #3: Yes

4. Is the manuscript presented in an intelligible fashion and written in standard English?

Reviewer #1: Yes

Reviewer #2: Yes

Reviewer #3: Yes

5. Review Comments to the Author

Reviewer #1: I have reviewed the manuscript entitled 'Predictive Value of Uric Acid to Albumin Ratio for Carotid Atherosclerosis in Type 2 Diabetes Mellitus: A retrospective study'.

The manuscript sounds scientific and is well-presented. However several issues should be evaluated in order to provide more precise deductions.

The title was written twice please check.

In the introduction section the role of uric acid/ albumin should be mentioned with recent investigations citing 'The Association of Serum Uric Acid/Albumin Ratio with No-Reflow in Patients with ST Elevation Myocardial Infarction', 'Predictive value of uric acid/albumin ratio for the prediction of new-onset atrial fibrillation in patients with ST-Elevation myocardial infarction' and 'Association of Uric Acid Albumin Ratio with Recurrence of Atrial Fibrillation after Cryoballoon Catheter Ablation'. These articles will emphasize the concomitancy of inflammation and uric acid/ albumin.

Please explain the inclusion and exclusion criteria more in detail.

Please emphasize the hypothesis of the study.

The role of albumin should also be mentioned since it has several predictive effects in addition to other parameters such as in Naples score, PNI and others. Please mention the importance of albumin citing 'Overlap Between Nutritional Indices in Patients with Acute Coronary Syndrome: A Focus on Albumin' and 'Evaluation of Naples Score for Long-Term Mortality in Patients With ST-Segment Elevation Myocardial Infarction Undergoing Primary Percutaneous Coronary Intervention'.

Reviewer #2: 1. The authors should consistently use either sex or Gender in every places.

2. In the table, "n" should be used instead of "N". “N” denotes population.

3. Kindly provide a more detailed description of Figures 1, 2 and 3.

4. The authors have not addressed the topic of women patients, and they have not even mentioned it in the limitations section.

Reviewer #3: After reviewing the manuscript, I would like to highlight several points as drawbacks

*The study’s sample size may limit the generalizability of its findings, as it is confined to a particular demographic.

*The retrospective nature of the study introduces potential biases and limits the capacity to infer causality.

As recommendations, details on how carotid IMT and plaques were measured could be clarified to ensure reproducibility. Additionally, addressing potential confounding variables in analyses would strengthen the conclusions drawn.

However, The study investigates the uric acid to albumin ratio (UAR) as a predictive biomarker for carotid atherosclerosis specifically in patients with type 2 diabetes mellitus (T2DM). This offers new insights, particularly since UAR combines elements of both nutritional status and inflammatory response.

6. PLOS authors have the option to publish the peer review history of their article (what does this mean? ). If published, this will include your full peer review and any attached files.

**Do you want your identity to be public for this peer review?** For information about this choice, including consent withdrawal, please see our Privacy Policy .

Reviewer #1: No

Reviewer #2: No

Reviewer #3: No

---

## [Author Response · Author response to Decision Letter 1]

16 Feb 2025

We appreciate the time and effort you and the reviewers dedicated to providing constructive feedback. Your insightful comments have greatly improved our manuscript.We have carefully considered all the comments and suggestions provided by you and the reviewers. All detailed responses are included in the response letter for your review.

My responses are as follows:

Academic Editor Comments and author responses:

1. comment: Please ensure that your manuscript meets PLOS ONE's style requirements, including those for file naming. The PLOS ONE style templates can be found at

and

1. Reply: Thank you for your reminder. I have carefully reviewed PLOS ONE's style requirements and ensured that my manuscript is formatted according to the guidelines provided.

2. Comment: Thank you for stating the following financial disclosure:

“This work was supported by the National Natural Science Foundation of China (Grant No. 82070862, No.82370840) and the National Key R&D Program of China (Grant No. 2020YFC2008900). These funds were obtained by Professor Jin.”

2. Reply: Thank you for your feedback regarding the financial disclosure. We recognize the importance of clarifying the role of the funders in our study. I have added the role of the funder, Professor Jin, in the Funding Statement section of the manuscript and have also refined the CRediT authorship contribution statement. As requested, we will include this amended Role of Funder statement in our cover letter. Thank you once again for your guidance.

3. Comment: In the online submission form, you indicated that [The datasets used and analyzed during the current study are available from the corresponding author upon reasonable request].

3. Reply: Thank you for your guidance regarding data availability. I will upload all data underlying the findings described in the manuscript as supplementary information (S1 Dataset). This will ensure that the data is freely accessible to other researchers, in compliance with PLOS journals' policies. Thank you again for your assistance.

4. Comment: Your ethics statement should only appear in the Methods section of your manuscript. If your ethics statement is written in any section besides the Methods, please move it to the Methods section and delete it from any other section. Please ensure that your ethics statement is included in your manuscript, as the ethics statement entered into the online submission form will not be published alongside your manuscript.

4. Reply: Thank you for your valuable feedback on the ethics statement placement. We have revised the manuscript to include the ethics statement only in the Methods section and removed it from all other sections to comply with your guidelines.

5. Comment: Please review your reference list to ensure that it is complete and correct. If you have cited papers that have been retracted, please include the rationale for doing so in the manuscript text, or remove these references and replace them with relevant current references. Any changes to the reference list should be mentioned in the rebuttal letter that accompanies your revised manuscript. If you need to cite a retracted article, indicate the article’s retracted status in the References list and also include a citation and full reference for the retraction notice.

5. Reply: Thank you very much for your valuable feedback regarding our reference list. We have thoroughly reviewed the references to ensure that they are complete and accurate. We do not cite any retracted papers, and all references have been updated using the PLOS ONE style in EndNote.

Responses to reviews

Reviewer # 1:

1. Comment: The manuscript sounds scientific and is well-presented.

1. Reply: Thank you very much indeed for your comments. We are heartened by your positive remarks on our manuscript, which have reinforced our belief in the value of our research.

2. Comment: The title was written twice please check.

2. Reply: Thank you for bringing the issue of the repeated title to our attention. I sincerely apologize for my carelessness. We have corrected this and ensured that the title appears only once in the manuscript. We appreciate your careful review.

3. Comment: In the introduction section the role of uric acid/ albumin should be mentioned with recent investigations citing 'The Association of Serum Uric Acid/Albumin Ratio with No-Reflow in Patients with ST Elevation Myocardial Infarction', 'Predictive value of uric acid/albumin ratio for the prediction of new-onset atrial fibrillation in patients with ST-Elevation myocardial infarction' and 'Association of Uric Acid Albumin Ratio with Recurrence of Atrial Fibrillation after Cryoballoon Catheter Ablation'. These articles will emphasize the concomitancy of inflammation and uric acid/ albumin.

3. Reply: Thank you very much indeed for your comments. We appreciate your suggestion regarding the role of uric acid/albumin in the introduction. In response, we have revised the introduction to include a discussion of recent studies you recommended in lines 91-96, highlighting the importance of the uric acid/albumin ratio and emphasizing its relationship with inflammation.

4. Comment: Please explain the inclusion and exclusion criteria more in detail.

4. Reply: Thank you very much for your comments. We appreciate your request for a more detailed explanation of the inclusion and exclusion criteria. In response, we have expanded the Study Population section of the manuscript to provide greater clarity on these criteria. Specifically, we have elaborated on the inclusion and exclusion criteria in lines 116-117 and 128-134.

5. Comment: Please emphasize the hypothesis of the study.

5. Reply: Thank you for your valuable suggestion to highlight the study's hypothesis. In response, we have revised the manuscript to present a clearer and more prominent statement of our hypothesis. This has been incorporated into the introduction section to ensure it stands out and effectively conveys the key objectives of our research. You can find the revised text in lines 103-106 of the introduction section.

6. Comment: The role of albumin should also be mentioned since it has several predictive effects in addition to other parameters such as in Naples score, PNI and others. Please mention the importance of albumin citing 'Overlap Between Nutritional Indices in Patients with Acute Coronary Syndrome: A Focus on Albumin' and 'Evaluation of Naples Score for Long-Term Mortality in Patients With ST-Segment Elevation Myocardial Infarction Undergoing Primary Percutaneous Coronary Intervention'.

6. Reply: Thank you very much indeed for your comments. Thank you for your insightful comment regarding the role of albumin. We recognize its significant predictive effects in various clinical parameters, including the Naples score and the Prognostic Nutritional Index (PNI). In response, we have revised the manuscript to include a discussion on the importance of albumin in lines 83-89 of the introduction.

Reviewer # 2:

1. Comment: The authors should consistently use either sex or Gender in every places.

1. Reply: Thank you for your valuable feedback regarding the use of terminology. We apologize for this oversight and recognize the importance of maintaining consistency in the manuscript. In response, we have reviewed the text and replaced all instances of "sex" with "gender" to ensure consistency throughout the manuscript. Thank you again for your constructive reminder.

2. Comment: In the table, "n" should be used instead of "N". “N” denotes population.

2. Reply: Thank you for your comment regarding the notation in the table. We appreciate your attention to detail. In response, we have revised the table to use "n" instead of "N," as "N" typically denotes the population. This change enhances the accuracy of our data presentation.

3. Comment: Kindly provide a more detailed description of Figures 1, 2 and 3.

3. Reply: Thank you for your constructive feedback regarding the figures. We appreciate your suggestion for more detailed descriptions. In response, we have enhanced the descriptions of Figures 1, 2, and 3 in the manuscript to provide clearer insights into the data presented.

4. Comment: The authors have not addressed the topic of women patients, and they have not even mentioned it in the limitations section.

4. Reply: Thank you for your insightful comment regarding the inclusion of female patients in our study. We acknowledge the importance of considering gender differences in clinical research. We have added the results of the gender-stratified analysis and discussed them in lines 420-426 of the discussion section in the Revised Manuscript with Track Changes (in lines 417-423 of the final Manuscript). And indeed, after adjusting for confounding factors, it was found that female patients with high UAR levels have a higher risk of disease. Thank you once again for your profound insights. Your input has helped us enhance the comprehensiveness of our manuscript.

Reviewer # 3:

1. Comment: Details on how carotid IMT and plaques were measured could be clarified to ensure reproducibility.

1. Reply: Thank you for your valuable comments regarding the measurement methods for carotid intima-media thickness (IMT) and plaques. We agree that a clear description of these methods is essential for ensuring reproducibility. In response, we have provided more detailed descriptions of the ultrasound probe frequency, patient positioning, and specific examination sites, which have been added to lines 155-159 and 162-166 of the Revised Manuscript with Track Changes (155-159 and 160-164 of the final Manuscript). We appreciate your feedback once again.

2. Comment: Addressing potential confounding variables in analyses would strengthen the conclusions drawn.

2. Reply: Thank you for your insightful comment regarding the consideration of potential confounding variables in our analyses. We recognize that addressing these variables is crucial for strengthening the validity of our conclusions. In response, we conducted stratified analyses based on gender and age, and we have discussed these findings in lines 420-426 and 440-445 of the Discussion section in the Revised Manuscript with Track Changes (in lines 417-423 and 437-442 of the final Manuscript). In addition, we have further explored this topic in the limitations section, incorporating your valuable comments.

3. Comment: However, the study investigates the uric acid to albumin ratio (UAR) as a predictive biomarker for carotid atherosclerosis specifically in patients with type 2 diabetes mellitus (T2DM). This offers new insights, particularly since UAR combines elements of both nutritional status and inflammatory response.

3. Reply: Thank you for your thoughtful comments. We greatly appreciate your recognition of our study on the uric acid to albumin ratio (UAR) as a predictive biomarker for carotid atherosclerosis (CAS) in type 2 diabetes mellitus (T2DM). Your insights regarding the interplay of nutritional status and inflammatory response are invaluable to us. Thank you for helping me refine my manuscript to a higher standard.

More detailed responses are included in the response letter. Thank you all once again for your constructive feedback, which undoubtedly helps enhance the quality and clarity of our manuscript.

Best wishes.

---

## [Decision Letter · Decision Letter 1]

25 Feb 2025

Predictive Value of Uric Acid to Albumin Ratio for Carotid Atherosclerosis in Type 2 Diabetes Mellitus: A Retrospective Study

PONE-D-24-50053R1

Dear Dr. Jin,

We’re pleased to inform you that your manuscript has been judged scientifically suitable for publication and will be formally accepted for publication once it meets all outstanding technical requirements.

Kind regards,

Aleksandra Klisic

Academic Editor

PLOS ONE

Additional Editor Comments (optional):

Reviewers' comments:

Reviewer's Responses to Questions

**Comments to the Author**

1. If the authors have adequately addressed your comments raised in a previous round of review and you feel that this manuscript is now acceptable for publication, you may indicate that here to bypass the “Comments to the Author” section, enter your conflict of interest statement in the “Confidential to Editor” section, and submit your "Accept" recommendation.

Reviewer #1: All comments have been addressed

Reviewer #3: All comments have been addressed

2. Is the manuscript technically sound, and do the data support the conclusions?

Reviewer #1: Yes

Reviewer #3: Yes

3. Has the statistical analysis been performed appropriately and rigorously? 

Reviewer #1: Yes

Reviewer #3: Yes

4. Have the authors made all data underlying the findings in their manuscript fully available?

Reviewer #1: Yes

Reviewer #3: Yes

5. Is the manuscript presented in an intelligible fashion and written in standard English?

Reviewer #1: Yes

Reviewer #3: Yes

6. Review Comments to the Author

Reviewer #1: I have reviewed the manuscript entitled ' Predictive Value of Uric Acid to Albumin Ratio for Carotid Atherosclerosis in Type 2 Diabetes Mellitus: A Retrospective Study'. The manuscript is acceptable after the revisions.

Reviewer #3: (No Response)

7. PLOS authors have the option to publish the peer review history of their article (what does this mean? ). If published, this will include your full peer review and any attached files.

**Do you want your identity to be public for this peer review?** For information about this choice, including consent withdrawal, please see our Privacy Policy .

Reviewer #1: No

Reviewer #3: No

---

## [Editor Report · Acceptance letter]

PONE-D-24-50053R1

PLOS ONE

Dear Dr. Jin,

I'm pleased to inform you that your manuscript has been deemed suitable for publication in PLOS ONE. Congratulations! Your manuscript is now being handed over to our production team.

Kind regards,

on behalf of

Dr. Aleksandra Klisic

Academic Editor

PLOS ONE